# Investigation of the optimal platinum-based regimen in the postoperative adjuvant chemotherapy setting for early-stage resected non-small lung cancer: a Bayesian network meta-analysis

Lan-Lan Pang  ,[1] Jia-Di Gan,[1] Yi-Hua Huang,[1] Jun Liao,[1] Yi Lv,[2] Wael Abdullah-Sultan Ali,[1] Li Zhang,[1] Wen-Feng Fang[1]

[1]Department of Medical Oncology, State Key Laboratory of Oncology in South China, Collaborative Innovation Center for Cancer Medicine, Sun Yat-sen University Cancer Center, Guangzhou, China
[2]Zhongshan School of Medicine, Sun Yat-Sen University, Guangzhou, China

**Correspondence to**
Dr Wen-Feng Fang;
fangwf@sysucc.org.cn

## ABSTRACTS

**Objective** This study aimed to compare the efficacy and safety of different platinum adjuvant chemotherapy regimens for early-stage resected non-small-cell lung cancer (NSCLC).

**Design** Systematic review with network meta-analysis of randomised trials.

**Data sources** PubMed, EMBASE, The Cochrane Library, Web of Science and Scopus Google Scholar were searched through 12 March 2021.

**Eligibility criteria** Eligible randomised controlled trials (RCTs) comparing the postoperative platinum chemotherapy regimen with the observation-controlled group or comparing two platinum chemotherapy regimens head-to-head were included.

**Data extraction and synthesis** The primary outcome was the efficacy of adjuvant chemotherapy regimens including relapse-free survival (RFS), overall survival (OS), 2-year, 3-year, 5-year RFS rate and OS rate. The secondary outcome was the rate of grade 3–4 toxicity assessments. Cochrane Handbook (V.5) was used for the risk of bias assessment. Analyses were performed using R software V.4.3.1.

**Results** 20 RCTs with a sample size of 5483 were enrolled in meta-analysis. The chemotherapy group had a significant RFS and OS advantage compared with the observation group (HR 0.67; 95% CI 0.56 to 0.81, p<0.0001; HR 0.80; 95% CI, 0.73 to 0.88, p<0.0001, respectively). Compared with the observation arm, only the 'cisplatin_vinorelbine' regimen had a significant RFS and OS advantage (HR 0.63; 95% CI 0.43 to 0.87; HR 0.74; 95% CI 0.63 to 0.87, respectively) while the remaining chemotherapy regimens had no significant difference of efficacy compared with the observation group. In terms of the safety of adjuvant chemotherapy, the incidence of haematological toxicities and nausea/vomiting was not significantly higher in the 'cisplatin_vinorelbine' arm than in other chemotherapy group.

**Conclusion** This study summarised the adjuvant cytotoxicity chemotherapy regimens for patients with early-stage resected NSCLC. Our analysis may provide

## STRENGTHS AND LIMITATIONS OF THIS STUDY

⇒ The Bayesian network meta-analysis provided access to directly compare the efficacy and safety of different adjuvant chemotherapy regimens for non-small-cell lung cancer patients.

⇒ It is difficult to determine the optimal subgroup populations who may obtain the benefit from certain cytotoxicity chemotherapy regimens.

⇒ We did not incorporate any targeted or immune-biological therapy into the analysis due to the open-loop of interventions.

⇒ This study could not present a subgroup analysis stratified by postoperative radiotherapy or not as it was undertaken according to every centre's policy.

some guiding significance for the clinicians when determining the optimal chemotherapy regimen.

## INTRODUCTION

Lung cancer is the most common cause of cancer-related deaths worldwide and patients with non-small lung cancer (NSCLC) account for almost 85% of all cases.[1 2] Complete surgical resection is recognised as the standard treatment for patients with early-stage NSCLC. However, 30%–70% of patients ultimately experience local or distant relapse and the 5-year survival rate ranges from only 30% to 60%.[3 4] The presence of micrometastases, which could not be detected by conventional diagnostic techniques, increases the rate of relapse and exerts a negative impact on the survival of patients.[5] Therefore, adjuvant postoperative systematic therapy is necessary for patients experiencing surgical resection.

A pooled analysis conducted by the Lung Adjuvant Cisplatin Evaluation (LACE)

Collaborative Group suggested that cisplatin-based adjuvant chemotherapy yield an absolute overall survival (OS) benefit of 5.40% at 5 years.[6] In light of this promising data, the American Society of Clinical Oncology guidelines recommended the cisplatin-based adjuvant postoperative chemotherapy regimen as the standard treatment for early-stage NSCLC patients.[7] However, the optimal adjuvant chemotherapy regimen has not yet been determined. 'Cisplatin_vinorelbine' has shown its promising efficacy compared with the observation group and is recognised as the standard adjuvant chemotherapy regimen currently.[8 9] In addition, several randomised controlled trials (RCTs) have also investigated the feasibility, efficacy and safety of cisplatin plus non-vinorelbine third-generation drug as the postoperative adjuvant chemotherapy.[10–12] Among them, 'cisplatin_pemetrexed' is characterised as having promising efficacy and an acceptable safety profile in comparison with 'cisplatin_vinorelbine'.[12 13] Thus, despite the lack of level 1 data concerning the efficacy of cisplatin plus vinorelbine, it was still recommended by the National Comprehensive Cancer Network guidelines as the optional postoperative adjuvant chemotherapy regimen for non-squamous NSCLC.[14] Meanwhile, although carboplatin-based adjuvant regimens have not yet been recommended by guidelines, it might be a favourable choice for patients with comorbidities and unsuited to receive cisplatin when accompanied by the late toxicity.[15 16]

Recent advances in individualised treatment based on molecular and biological profiling have shaped the future of postoperative adjuvant chemotherapy. Substantial numbers of patients benefit from the corresponding adjuvant regimen. The patients unsuited to adjuvant cytotoxicity chemotherapy may benefit from the first-generation tyrosine kinase inhibitor (TKI) but serval clinical trials showed poor outcomes of TKI adjuvant therapy until the advent of ADJUVANT study.[17–19] However, individualisation could not apply to all NSCLC patients and those patients inapplicable to it have no choice but receive platinum adjuvant chemotherapy.

Postoperative adjuvant chemotherapy setting for early-stage resected NSCLC is of great significance for the oncological community in routine clinical practice. However, it seems difficult to conduct large RCT to figure out the preferred platinum chemotherapy regimen and using the published data might be an alternative option. Hence, we conducted this systematic review and network meta-analysis (NMA) aiming to compare the efficacy and safety of different platinum adjuvant chemotherapy regimens.

## METHODS
This systematic review and NMA was performed in accordance with the Preferred Reporting Items for Systematic Reviews and Meta-Analyses guidelines for RCTs (The PRISMA statement) (online supplemental material 1).

## Data sources
Two authors (LP and JG) independently searched the records in the electronic database of PubMed, EMBASE, and The Cochrane Library, Web of Science and Scopus Google Scholar. The searching terminal date was 12 March 2021. Searching terms were focused on NSCLC, adjuvant chemotherapy and platinum. If necessary, an additional manual search of related literature in the reference list would be carried out to enrol any relevant publications. The datasets used in this analysis could be obtained from the corresponding author on request. Records were imported into Endnote V.X9 software to eliminate duplications. The detailed strategy was presented in online supplemental material 2.

## Trial selection criteria and trial identification
Two authors (LP and Y-HH) independently reviewed the title, abstracts and keywords of identified citations to select appropriate articles for a full review. Any disagreement was resolved by consensus or through the judgement of a senior author (WF). Trials would be eligible only if meeting all the following criteria: (1) patients with completely resected NSCLC (squamous and non-squamous) at stage IB–IIIA; (2) Eligible RCTs comparing the postoperative platinum chemotherapy regimen with observation-controlled group or those concerning two platinum chemotherapy regimens head-to-head comparison; given that vinorelbine, etoposide, pemetrexed, docetaxel, paclitaxel, gemcitabine, vindesine are currently commonly used in the routine clinical practice, the counterpart of the platinum doublet including these above-mentioned drugs were considered eligible; and a platinum triplet must be a platinum doublet combined with anti-angiogenesis drug; (3) full-text publications or conference abstract and (4) non-language restrictions. Publications would be disregarded if meeting any of the following criteria: (1) any single perioperative chemotherapy, neoadjuvant chemotherapy, maintenance chemotherapy or radiochemotherapy; (2) chemonaive patients with advanced, incurable or recurrent NSCLC and (3) no RCTs including reviews, case series or retrospective trials.

## Outcomes and data extraction
Two authors (LP and JL) independently performed data extraction and any discrepancies were eliminated by consensus. Data for eligible trials related to basic characteristics were extracted, including: trial/author name, publication year, country of origin/multi-centres, sample size per group, chemotherapy regimen, administration dose of each treatment and number of cycles, follow-up time, phases II/III, basic patients' characteristics (age, sex ratio, stage, pathology, performance status).

The primary outcome was the efficacy of adjuvant chemotherapy regimens including relapse-free survival (RFS), OS, 2-year, 3-year, 5-year RFS rate and OS rate. RFS was defined as the interval from day of randomised assignment to disease recurrence or death, whichever occurred

first, while OS referred to the time from randomisation to death from any cause.

The secondary outcomes were the rate of grade 3–4 toxicity assessment, including haematological (anaemia, neutropaenia, febrile neutropaenia and thrombocytopaenia) and non-haematological (nausea/vomiting) adverse events. No additional information was requested from the authors. Also, Parmar's method would be used to obtain survival if necessary.

### Quality and risk of bias assessment
Two researchers (LP and JG) independently assessed the risk of bias of enrolled trials according to the recommendations of the Cochrane Handbook of Systematic Reviews of Interventions (V.5, http://handbook.cochrane.org).[20] Disagreements were resolved by discussion.

### Data synthesis and analysis
For RFS and OS, the logarithm of HR and their SE were pooled into analysis through a Bayesian multiple treatment NMA with random effects. As for the dichotomous variables, OR with 95% CI was applied to calculate. When a network diagram indicated two or more independent loops, only the loop containing 'cisplatin_vinorelbine' was selected for further analysis.

Random effects and consistency model was computed using Markov chain Monte Carlo methods with Gibbs sampling based on simulations of 50 000 iterations and 20 000 adaptions in each of 4 chains. For a forest plot, 'observation' was chosen as the common reference comparator in the analysis of RFS and OS while 'cisplatin_vinorelbine' was applied in the analysis of chemotherapy toxicity. A league table for the survival analysis was presented with the logarithm of HR and their 95% CI. The surface under the cumulative ranking curve (SUCRA), represents the percentage of efficacy achieved by an agent compared with an imaginary agent that is always the best without uncertainty (ie, SUCRA=100%).[21] Namely, the SUCRA value would be 1 if treatment is certain to be the best and 0 if it is certain to be the worst. Higher SUCRA scores correspond to a higher ranking for extending survival.

All analyses in this article were performed using R software V.4.3.1 and the gemtc package version 0.8 that interfaces with JAGS V.4.3.0 for computing a Markov chain.

### Patient and public involvement
Patients and the public were not involved in the design or planning of the study.

## RESULTS
### The search process, study characteristics and quality assessment
A total of 3432 records yielded after searching the database and 1991 records were left to be identified after removing 1692 duplications. Then, 1905 irrelevant citations were eliminated by skimming their title, abstracts

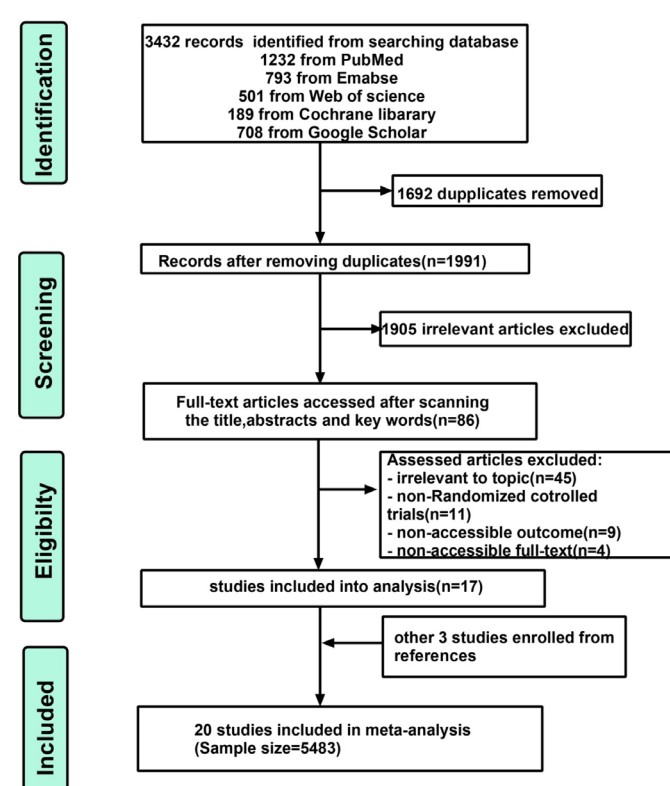

**Figure 1** Flow diagram of enrolled studies selection.

and keywords, leaving 86 articles to be considered potentially eligible. Sixty-nine articles were further excluded for irrelevant to topic (n=45); non-RCTs (n=11); non-accessible outcome (n=9); non-accessible text (n=4) and 17 articles were enrolled into analysis after skimming their full text. Additionally, three studies were included by browsing the references. Finally, 20 studies with a sample size of 5483 were enrolled into meta-analysis (figure 1).

Table 1 shows the basic characteristics of enrolled studies and participants. These studies were conducted in countries worldwide, most of which were Japan[22–26] and China,[27–29] [30] followed by France,[31 32] Germany,[33] Italy,[34] USA,[35] UK[36] and North Carolina.[10] Noteworthy, three of them were conducted in multicountries.[37–39] Eleven studies compared the postoperative platinum chemotherapy regimen with the observation-controlled group while the remaining nine studies comparing two platinum chemotherapy regimens head-to-head. Chemotherapy regimens included 'cisplatin_vinorelbine', 'cisplatin_pemetrexed', 'cisplatin_gemcitabine', 'cisplatin_vindesine', 'cisplatin_etoposide', 'cisplatin _docetaxel', 'carboplatin_gemcitabine', 'carboplatin_pemetrexed', 'carboplatin_paclitaxel', 'carboplatin_docetaxel_endostar', 'carboplatin_docetaxel', 'cisplatin _vinorelbine_endostar'. The median follow-up time ranged from 20.2 to 116 months. Of note, we have noticed four well-designed RCTs during the process of eligible trial selection.[40–43] Nevertheless, we could not obtain the specific outcomes in the subgroup populations per treatment regimens from these clinical trials, whereas only the outcomes concerning the comparison between the

**Table 1** Basic characteristics of enrolled studies and participants

| Trial | Publication year | Country of region | Chemotherapy regimes | Sample size | Median follow-up (months) | Phase | Radiotherapy | Patients characteristics (Pathology; stage; PS) | Age | Sex |
|---|---|---|---|---|---|---|---|---|---|---|
| ANITA[37] | 2006 | Multicentre | Cisplatin 100 mg/m² + vinorelbine 30 mg/m² vs observation | 840 (407 vs 433) | 76 | III | Not mandatory. | NSCLC; stage IB–IIIA | 59 (32–75) vs 59 (18–75) | 85% vs 87% |
| Big-lung-trial[36] | 2003 | UK | Cisplatin 80 mg/m² + vinorelbine 30mg/m² vs observation | 66 (37 vs 29) | 58.8 | - | Not mandatory | NSCLC; stage I–III | / | / |
| CALGB 9633[10] | 2004 | North Carolina | Carboplatin (AUC-6) +paclitaxel 200mg/m² vs observation | 344 (173 vs 171) | 74 | III | Not mentioned | NSCLC; T2 with pathologically negative lymph nodes | 61 (34–78) vs 62 (40–81) (range) | 65% vs 63% |
| CSLC0201[28] | 2016 | China | Carboplatin +docetaxel vs observation | 82 (43 vs 39) | 132 | - | Not mentioned | NSCLC; stage IB–IIIA | / | / |
| NATCH[39] | 2010 | Multi Centre | Carboplatin (AUC-6.0) +paclitaxel 200mg/m² vs observation | 423 (211 vs 212) | 51 | III | Postoperative radiotherapy was allowed in patients with pathologic N2 disease. | NSCLC; stage IA with tumour size more than 2cm, IB, II or T3N1 | 64 (33–81) vs 64 (36–89) (range) | 86% vs 88% |
| Barlesi et al[31] | 2015 | France | Cisplatin 75 mg/m² + gemcitabine 1250mg/m² vs cisplatin 75 mg/m² + docetaxel 75 mg/m² | 136 (67 vs 69) | 20.2 | - | Not mandatory. | NSCLC; stage IB–III | 57 (44–74) vs 57 (36–71) | 75% vs 74% |
| HOT0703[22] | 2020 | Japan | Cisplatin 40 mg/m² + gemcitabine 1000mg/m² vs carboplatin (AUC-5) + gemcitabine 1000 mg/m² | 102 (51 vs 51) | 69.6 | II | Not mentioned | NSCLC; stage IB–IIIA | 63 (40–72) vs 64 (36–74) (range) | 67% vs 63% |
| IALT[32] | 2004 | France | Cisplatin 100 mg/m² + vinorelbine 30 mg/m² vs observation | 500 (248 vs 262) | 49.2 | - | Not allowed. | NSCLC; stage I, II or III | / | / |
| JBR10[35] | 2005;2010 | North American | Cisplatin + vinorelbine vs observation | 482 (240 vs 242) | 111.6 | III | Not mentioned | NSCLC; stage IB (T2N0) or II (T1-2N1) | 61 (35–82) vs 60.5 (34–78) | 64% vs 66% |
| JCOG9304[23] | 2003 | Japan | Cisplatin 80 mg/m² + vindesine 3 mg/m² vs observation | 159 (59 vs 60) | - | - | Not mentioned | NSCLC;N2 | 62 (41–75) vs62 (43–74) | 68% vs 62% |
| Jing Wang et al[29] | 2012 | China | Cisplatin 80 mg/m² + vinorelbine 30 mg/m² vs observation | 451 (225 vs 226) | 46 | - | Not mentioned | NSCLC; stage I, II and IIIA | 55 (38–83) vs 58 (38–82) | 71% vs 75% |
| JIPANG[24] | 2020 | Japan | Cisplatin 75 mg/m² + pemetrexed 500mg/m² vs cisplatin 80 mg/m² + vinorelbine 25mg/m² | 804 (402 vs 402) | 45.2 | III | Not allowed | Non-squamous NSCLC; N2 stage II or IIIA | 65 (58–69) vs 64 (57–67) | 60% vs 58% |
| Roselli et al[34] | 2006 | Italy | Cisplatin 100 mg/m² + etoposide 120mg/m² vs observation | 140 (70 vs 70) | 40.31 | - | Not mentioned | NSCLC; stage IB disease (pT2N0) | 64.7±9.9 vs 62.9±9.2 | 91% vs 76% |
| JLCSSG[25] | 1993 | Japan | Cisplatin 80 mg/m² + vindesine 3mg/m² vs observation | 181 (90 vs 91) | 31.2 | - | Not mentioned | NSCLC; stage III | 56.3±9.1 vs 58.9±8.4 | 77%S. 87% |
| Chen et al[11] | 2015 | China | Cisplatin 75 mg/m² + docetaxel 75mg/m² vs cisplatin 75 mg/m² + gemcitabine 1250 mg/m² | 92 (45 vs 47) | 22 | - | Not mentioned | NSCLC; stage II–III | 55 (32–67) vs 56 (31–67) (range) | 84% vs 87% |
| Schmid-Bindert et al[38] | 2015 | Germany, France, and Spain | Cisplatin 75 mg/m² + pemetrexed 500mg/m² vs carboplatin (AUC-5) + pemetrexed 500mg/m² | 112 (63 vs 59) | - | II | Not-mentioned | NSCLC; stage IB, IIA or IIB | 61 (44–75) vs 59 (43–69) | 78% vs 70% |
| TORG0503[26] | 2019 | Japan | Cisplatin 80 mg/m² + docetaxel 60mg/m² vs carboplatin AUC-6+Paclitaxel 200mg/m2 | 111 (58 vs 53) | - | II | Not mentioned | NSCLC; stage IB, II and IIIA | 63 (33–70) vs 59 (34–70) | 60% vs 66% |

Continued

**Table 1** Continued

| Trial | Publication year | Country of region | Chemotherapy regimes | Sample size | Median follow-up (months) | Phase | Radiotherapy | Patients characteristics (Pathology; stage, PS) | Age | Sex |
|---|---|---|---|---|---|---|---|---|---|---|
| TREAT[33] | 2015 | German | Cisplatin 50 mg/m² + vinorelbine 25mg/m² vs cisplatin 75 mg/m² + pemetrexed 500mg/m² | 132 (67 vs 65) | 36 m | II | Not allowed | NSCLC; stages IB, IIA, IIB, | 58 (40–73) vs 60 (38–74) | 72%vs 77% |
| Yanzhuo Yang et al[30] | 2012 | China | Carboplatin (AUC=5–6) + docetaxel 75 mg/m²+ endostar 15 mg vs carboplatin+docetaxel | 76 (38 vs 38) | 22 | - | Not mentioned | NSCLC; stage IB-III | 55.6 (36–74) vs 60.2 (45–77) | 31% vs 27% |
| Chen et al[27] | 2017 | China | Cisplatin 75 mg/m² + vinorelbine 25mg/m²+ endostar 7.5 mg/m² vs cisplatin+vinorelbine | 250 (125 vs 125) | 60 | III | Not mentioned | NSCLC; stage IB to IIIA | 58 (33–75) vs 55.5 (37–71) | 66% vs 67% |

AUC, area under the curve; NSCLC, non-small cell lung cancer; PS, performance status.

observation group and 'cisplatin _vinorelbine' were available in "BLT"(Big-lung-trial) or "IALT"(International Adjuvant Lung Trial) trial.[32 36] Given that the objective of the present work was to investigate the optimal platinum-based regimen in the postoperative adjuvant chemotherapy setting for early-stage resected NSCLC, we only enrolled the BLT and IALT while leaving the remaining four RCTs out.

Based on the Cochrane Risk of Bias evaluation, one of the enrolled studies had a high risk of selection bias due to its quasi-randomised setting while nearly half the studies did not clarify the randomisation methods in detail. One study had a high risk of performance bias while nearly all of the studies did not mention using the blind design or not. Attrition and reporting bias did not exist in the enrolled studies except one whose bias could not be evaluated due to a conference abstract (online supplemental figure 1A,B).

### Primary outcome: RFS

Ten studies were enrolled to analyse the RFS of platinum postoperative adjuvant chemotherapy compared with the observation group. We found that the chemotherapy group had a significant RFS advantage compared vs the observation group (HR 0.67; 95% CI 0.56 to 0.81; $I^2$=64%, p<0.0001) (figure 2A). Moreover, in terms of 2 years RFS rate, 3 years RFS rate and 5 years RFS rate, the chemotherapy arm presented higher RFS rate in comparison with the observation arm (OR 1.62, 95% CI 1.20 to 2.18, $I^2$=68%, p=0.0016; OR= 1.50, 95% CI 1.11 to 2.03, $I^2$=72%, p=0.0081; OR 1.49, 95% CI 1.09 to 2.04, $I^2$=69%, p=0.0132; respectively) (figure 2B–D).

Fifteen clinical trials were enrolled to investigate the RFS of several chemotherapy regimens for patients with early-stage resected NSCLC. In the network evidence figure, each node represents a type of treatment. Solid lines connect treatments that are directly compared in at least one study. The thickness of connections varies based on the number of studies involved in a comparison. Network evidence of the comparisons for the best adjuvant chemotherapy concerning the RFS was shown in figure 3A. Almost all platinum chemotherapy has a direct comparison with 'observation' except 'cisplatin_docetaxel', 'cisplatin_pemetrexed', 'cisplatin_vinorelbine_endostar' and 'carboplatin_docetaxel_endostar'. Furthermore, the number of trials that compared 'cisplatin_vinorelbine' vs 'observation' ranked the first. Furthermore, compared with the observation arm, only the 'cisplatin_vinorelbine' regimen had a significant RFS advantage (HR 0.63; 95% CI 0.43 to 0.87) while there was no significant difference between the remaining chemotherapy regimens and the observation group-'carboplatin_docetaxel' (HR 0.58; 95% CI 0.24 to 1.4); 'carboplatin_docetaxel_endostar' (HR 0.42; 95% CI 0.05 to 3.8); 'carboplatin_paclitaxel' (HR 0.88; 95% CI 0.51 to 1.50); 'cisplatin_docetaxel' (HR 1.40; 95% CI 0.49 to 3.90); 'cisplatin_etoposide' (HR 0.53; 95% CI 0.23 to 1.30); 'carboplatin_pemetrexed' (HR 0.62; 95% CI 0.29 to 1.20); 'cisplatin_vindesine'

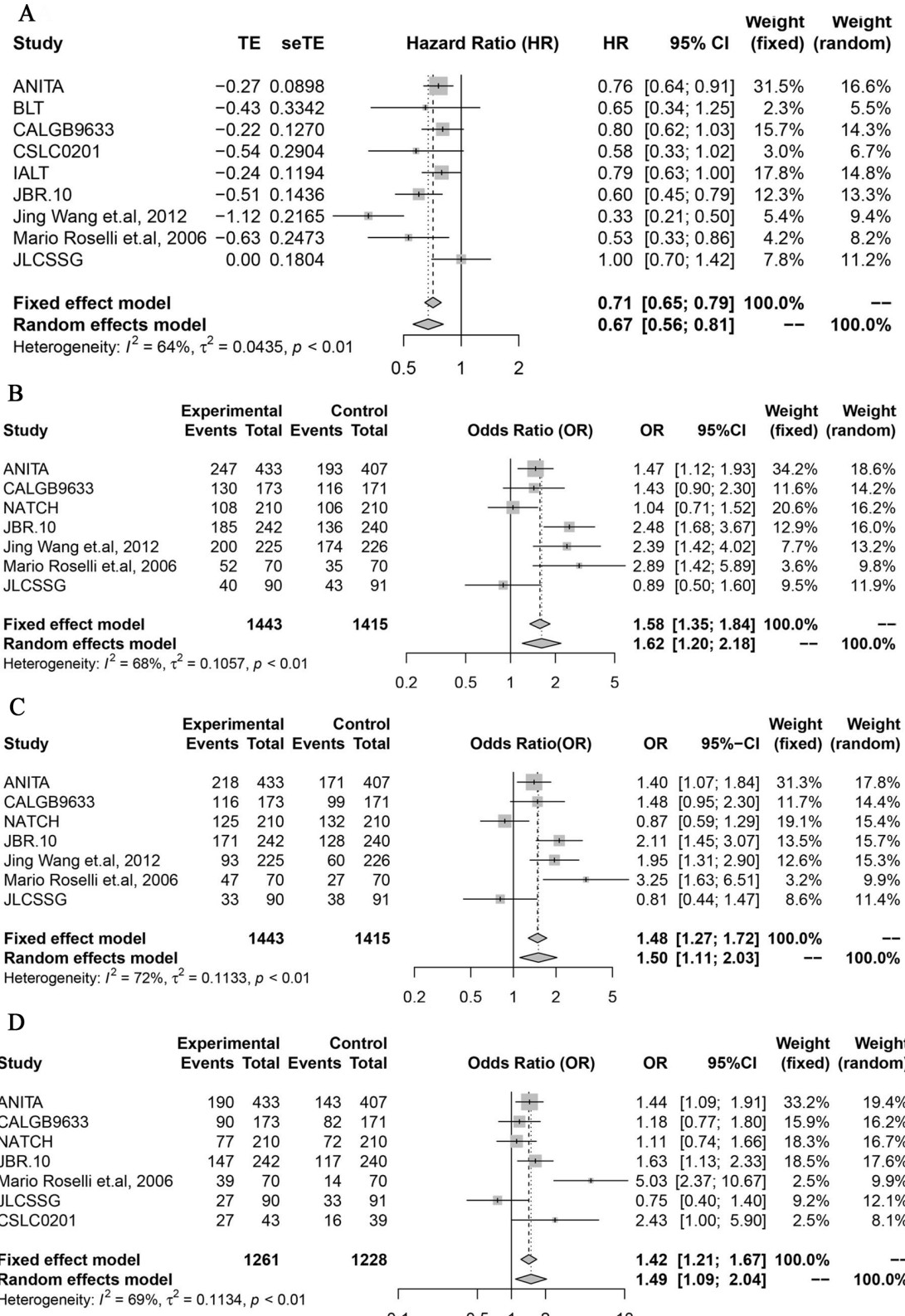

**Figure 2** (A) The efficacy of platinum-based postoperative adjuvant chemotherapy in improving the RFS compared with the observation group. (B) A 2-year RFS rate of the chemotherapy arm in comparison with the observation arm. (C) A 3-year RFS rate of the chemotherapy arm in comparison with the observation arm. (D) A 5-year RFS rate of the chemotherapy arm in comparison with the observation arm. RFS, relapse-free survival.

(HR 1.00; 95% CI 0.45 to 2.20); 'cisplatin_vinorelbine_endostar' (HR 0.60; 95% CI 0.24 to 1.40) (figure 3B). In addition, according to the league table, no significant difference was observed in the remaining comparisons, for example, 'cisplatin_vinorelbine' and 'cisplatin_pemetrexmed'. (figure 3C). In the rank of NMA, the biggest

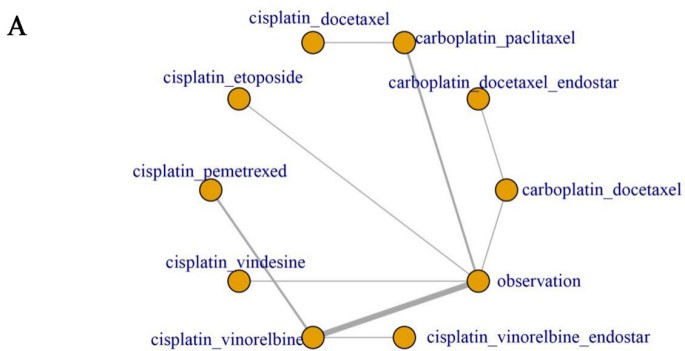

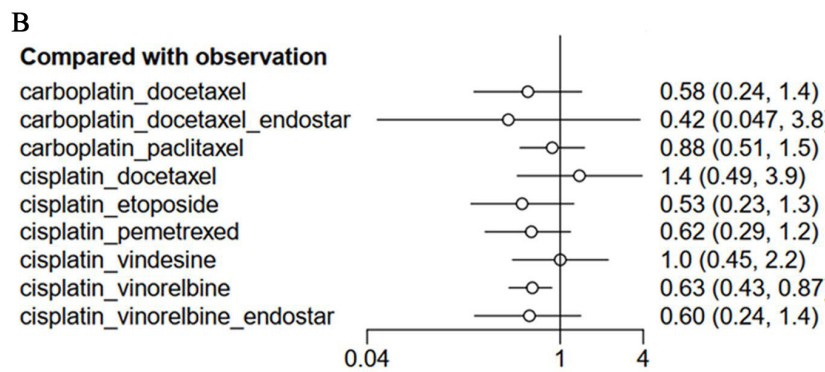

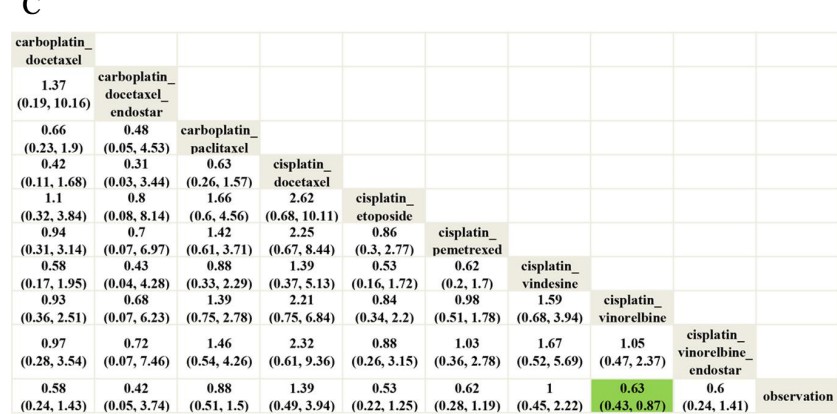

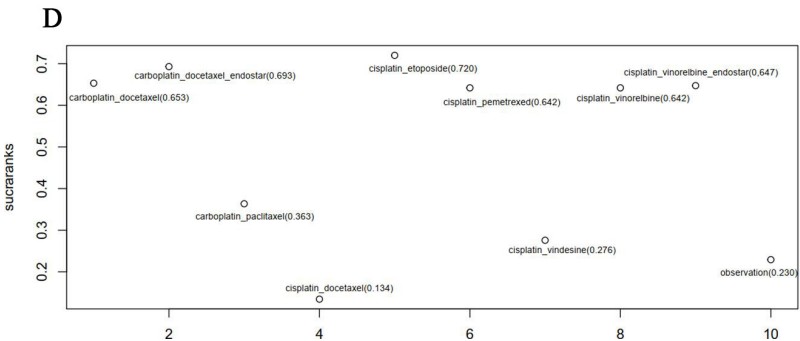

**Figure 3** (A) Network evidence of the comparisons for the best adjuvant chemotherapy concerning RFS. In the network evidence figure, each node represents a type of treatment. Solid lines connect treatments that are directly compared in at least one study. The thickness of connections varies based on the number of studies involved in a comparison. (B) Forest plots of the comparisons for the different cytotoxicity chemotherapy regimens concerning RFS. (C) The league table of the comparisons for the different cytotoxicity chemotherapy regimens concerning RFS. (D) SUCRA value of every cytotoxicity chemotherapy regimens concerning RFS. RFS, relapse-free survival; SUCRA, surface under the cumulative ranking curve.

SUCRA value holds the potential to be the best chemotherapy regimen in extending RFS. Our results showed that 'cisplatin_etoposide' (72.0%) was the most effective chemotherapy regime to improve RFS, followed by 'carboplatin_docetaxel_endostar' (69.30%), 'carboplatin_docetaxel' (65.31%), 'cisplatin_vinorelbine' (64.73%), 'cisplatin_pemetrexed' (64.2%), 'cisplatin_vindesine' (64.2%), 'carboplatin_paclitaxel' (36.3%), 'carboplatin_vindesine' (27.6%), 'cisplatin_vinorelbine_endostar' (23.0%) and 'cisplatin_docetaxel' (13.4%) (figure 3D). In terms of the 2-year RFS rate, 3-year RFS rate and 5-year RFS rate, 'cisplatin_vinorelbine' was an effective therapeutic method in improving 2-year RFS rate (OR 2.00, 95% CI 1.20 to 3.50); 'carboplatin_docetaxel' (OR 6.70, 95% CI 1.70 to 31.00) 'cisplatin_etoposide' (OR 3.30, 95% CI 1.20 to 9.10) and 'cisplatin_vinorelbine' (OR 1.70, 95% CI 1.10 to 2.90) had a significantly higher 3-year RFS rate than observation arm; wherase 'cisplatin_etoposide' (OR 5.10, 95% CI 1.70 to 16.00) was an effective method in improving 5-year RFS rate. However, there was no significant difference between the remaining chemotherapy regimens and observation arm concerning the RFS rate (online supplemental figure 2A–C).

### Primary outcome: OS

To compare the efficacy of platinum chemotherapy arm with the observation arm in extending OS, we enrolled 10 studies into analysis. Our results showed that the OS in the chemotherapy group was significantly longer than the observation group (HR 0.80; 95% CI 0.73 to 0.88; $I^2$=11%, p<0.0001) (figure 4A). As for 2-year OS rate, 3-year OS rate and 5-year OS rate, the chemotherapy group showed a significant survival advantage than the observation group (OR 1.31, 95% CI 1.06 to 1.61, $I^2$=21%, p=0.0117; OR 2.24, 95% CI 1.13 to 4.45, $I^2$=94%, p=0.0081; OR 1.36, 95% CI 1.08 to 1.72, $I^2$=54%, p=0.0100; respectively) (figure 4B–D).

Moreover, 14 clinical trials with 8 treatment regimens were enrolled to investigate the optimum adjuvant chemotherapy regimens in extending OS. Network evidence of these chemotherapy regimens was shown in figure 5A. Almost all platinum chemotherapy has a direct comparison with 'observation' except 'cisplatin_docetaxel', 'cisplatin_pemetrexed' and 'cisplatin_vinorelbine_endostar'. Similar to RFS, the number of trials related to 'cisplatin_vinorelbine' comparing versus 'observation' also ranked the first in extending OS. Compared with the observation arm, only the 'cisplatin_vinorelbine' regimen had a significant OS advantage (HR 0.74; 95% CI 0.63 to 0.87) while there was no significant difference between the remaining chemotherapy regimens and the observation group—'carboplatin_paclitaxel' (HR 0.90; 95% CI 0.70 to 1.20); 'cisplatin_docetaxel' (HR 1.10; 95% CI 0.50 to 2.30); 'cisplatin_etoposide' (HR 0.80; 95% CI 0.44 to 1.40); 'cisplatin_pemetrexed' (HR 0.73; 95% CI 0.47 to 1.10); 'cisplatin_vindesine' (HR 1.00; 95% CI 0.69 to 1.40); 'cisplatin_vinorelbine_endostar' (HR 0.75; 95% CI 0.46 to 1.20) (figure 5B). Similar to RFS, as the league

table showed, there was no significant difference among these remaining paltimum chemotherapy regimens in improving OS for patients with early-stage resected NSCLC (figure 5C). In terms of the rank of NMA, we found that 'cisplatin_vinorelbine' (74.8%) was the most effective therapeutic arm for the improvement of OS, followed by 'cisplatin_pemetrexed' (74.1%), 'cisplatin_vinorelbine_endostar' (68.5%), 'cisplatin_etoposide' (59.2%), 'carboplatin_paclitaxel' (43.6%), 'cisplatin_vindesine' (28.5%), 'cisplatin_docetaxel' (27.0%) and 'observation' (24.3%) (figure 5D). In addition, 'cisplatin_etoposide' was an effective chemotherapy regimen to improve the 2-year OS rate (OR 2.90, 95% CI 1.10 to 8.30) and 5-year OS rate (OR 6.70, 95% CI 1.70 to 3.10) while 'cisplatin_vinorelbine' had a significantly higher 5-year OS rate than observation arm (OR 1.70, 95% CI 1.10 to 2.90). However, there was no significant difference between the remain chemotherapy regimens and observation arm concerning 2-year OS rate, 3-year OS rate and 5-year OS rate (online supplemental figure 3A–C).

### Secondary outcome: anaemia, neutropaenia, thrombocytopaenia, febrile neutropaenia and nausea/vomiting

Fifteen clinical trials were enrolled to compare the presence of anaemia among different adjuvant chemotherapy regimens. We found that the incidence of anaemia in the 'carboplatin_pemetrexed' arm was significantly higher than the 'cisplatin_vinorelbine' arm (Log OR=44.06, 95% CI 0.21 to 151.05). However, there was no significant difference between the remaining chemotherapy regimens and 'cisplatin_vinorelbine' (figure 6A).

As for neutropaenia, we enrolled 13 clinical trials to compare the safety of different adjuvant chemotherapy regimens. The neutropaenia was more frequently observed in patients receiving 'carboplatin_gemcitabine' (log OR 60.60, 95% CI 0.94 to 193.7), 'cisplatin_gemcitabine' (log OR 60.11, 95% CI 0.77 to 193.27) and 'cisplatin_docetaxel' (log OR 59.71, 95% CI 0.56 to 192.89) than cisplatin_vinorelbine'. However, the incidence of neutropaenia in the 'cisplatin_vinorelbine' group was not significantly higher than the remaining chemotherapy regimens group (figure 6B).

Also, as for thrombocytopaenia, we enrolled 11 clinical trials to investigate the safety of different adjuvant chemotherapy regimens. Patients medicated with 'carboplatin_pemetrexed' were significantly observed thrombocytopaenia compared with 'cisplatin_vinorelbine' (log OR 36.43, 95% CI 2.87 to 114.27). However, there was no significant difference between the remaining chemotherapy regimens and 'cisplatin_vinorelbine' (figure 6C).

In terms of febrile neutropaenia, six clinical trials were included. Our results showed that there was no significant difference between these chemotherapy regimens and 'cisplatin_vinorelbine' (figure 6D).

In addition, 14 clinical trials were enrolled to analyse the observation of nausea/vomiting in NSCLC patients receiving postoperative adjuvant chemotherapy. We found that compared with 'cisplatin_vinorelbine',

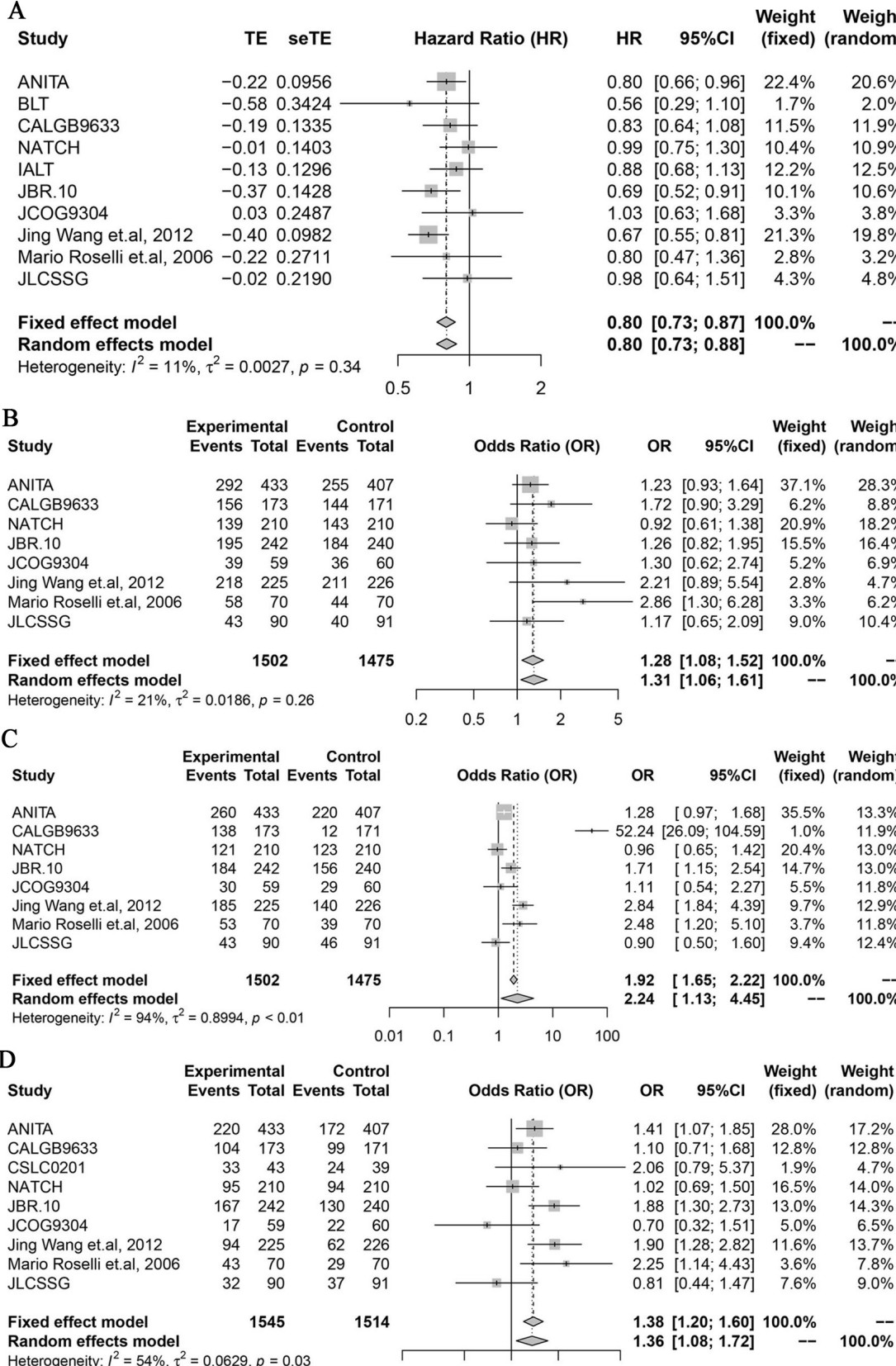

**Figure 4** (A) The efficacy of platinum-based postoperative adjuvant chemotherapy in improving the OS compared with the observation group. (B) A 2-year OS rate of the chemotherapy arm compared with the observation arm. (C) A 3-ear OS rate of the chemotherapy arm compared with the observation arm. (D) A 5-year OS rate of the chemotherapy arm compared with the observation arm. OS, overall survival.

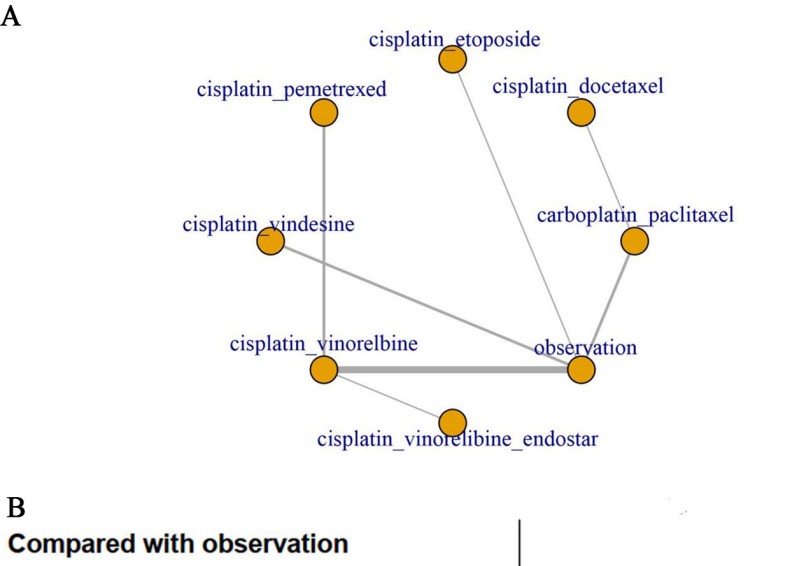

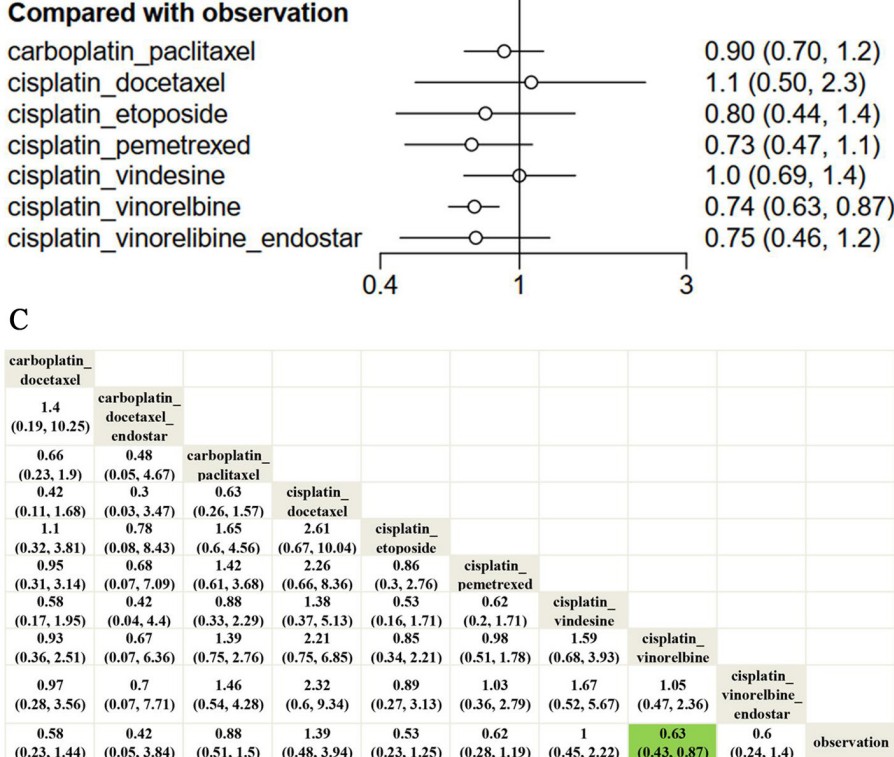

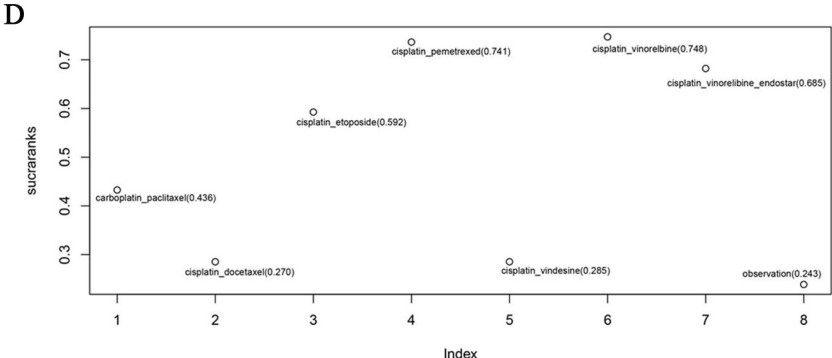

**Figure 5** (A) Network evidence of the comparisons for the best adjuvant chemotherapy concerning OS. (B) Forest plots of the comparisons for the different cytotoxicity chemotherapy regimens concerning OS. (C) The league table of the comparisons for the different cytotoxicity chemotherapy regimens concerning OS. (D) SUCRA value of every cytotoxicity chemotherapy regimens concerning OS. OS, overall survival; SUCRA, surface under the cumulative ranking curve.

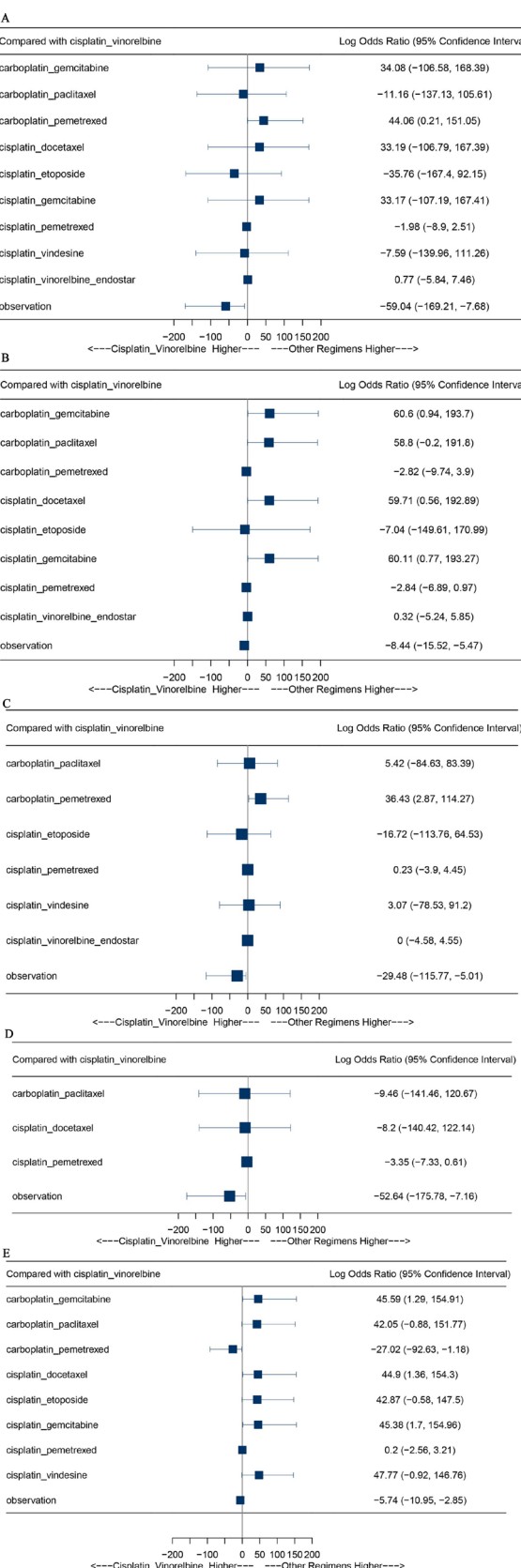

**Figure 6** (A) Forest plots of the comparisons for the different cytotoxicity chemotherapy regimens concerning the presence of anaemia. (B) Forest plots of the comparisons for the different cytotoxicity chemotherapy regimens concerning the presence of neutropaenia. (C) Forest plots of the comparisons for the different cytotoxicity chemotherapy regimens concerning the presence of thrombocytopaenia. (D) Forest plots of the comparisons for the different cytotoxicity chemotherapy regimens concerning the presence of febrile neutropaenia. (E) Forest plots of the comparisons for the different cytotoxicity chemotherapy regimens concerning the presence of nausea/vomiting.

'carboplatin_gemcitabine' (log OR 45.59, 95% CI 1.29 to 154.91), 'cisplatin_docetaxel' (log OR 44.90, 95% CI 1.36 to 154.30), 'cisplatin_gemcitabine' (log OR 45.38, 95% CI 1.70 to 154.96) had a higher incidence of nausea/vomiting. However, the nausea/vomiting was more frequently observed in patients receiving 'cisplatin_vinorelbine' than 'carboplatin_pemetrexed' (log OR −27.02, 95% CI −92.63 to −1.18). Furthermore, there was no significant difference between the remaining chemotherapy regimens and 'cisplatin_vinorelbine' (figure 6E).

## DISCUSSION

Recent advances in precision medicine have shaped the future of adjuvant therapy for patients with completely resected NSCLC.[19] However, for patients not applicable to targeted therapy or immunotherapy, cytotoxicity chemotherapy is still recommended by guidelines as the standard of care. It is of great significance to investigate the best platinum regimen in the postoperative adjuvant chemotherapy setting for early-stage NSCLC. Our results showed that adjuvant postoperative chemotherapy was beneficial to improve the RFS and OS compared with the observation arm. Among these chemotherapy regimens, the 'cisplatin_vinorelbine' arm was an effective therapeutic method to improve survival with tolerable toxicity; wherase the 'cisplatin_pemetrexed' arm did not show advantage over the other therapeutic methods in improving survival. Also, the 'cisplatin_etoposide' arm contributed to improving the RFS and OS rate for patients with resected NSCLC.

A meta-analysis, published in 1995, demonstrated that postoperative chemotherapy had a significant advantage in 5-year survival compared with the observation group, which opened up a new era of postoperative chemotherapy.[44] Of note, an updated analysis published in 2010 further confirmed the clinical value of postoperative adjuvant chemotherapy, with or without radiotherapy, for patients with operable NSCLC.[45] Moreover, a pooled analysis also indicated the promising role of cisplatin-based adjuvant chemotherapy on the 5-year survival.[6] Based on these findings, it is universally acknowledged that adjuvant cytotoxicity chemotherapy integrating platinum and third-generation agents is conducive to the improvement of postsurgical survival for NSCLC patients. In this study, the survival benefit was also observed in the platinum adjuvant chemotherapy setting. The adjuvant chemotherapy regimens were not limited to cisplatin-based chemotherapy, which extended to cisplatin/carboplatin in combination with other third-generation agents. Furthermore, the peak recurrence of patients with early-stage NSCLC is 2 years later after surgery and a higher 5-year RFS rate represented the lower possibility of postoperative relapse. In our study, not only 5-year RFS and OS benefit but the improvement of RFS, OS, 2-year RFS and OS rate, and 3-year RFS and OS rate were observed in the adjuvant chemotherapy group. These significant survival advantages could be explained in part by the fact that postoperative adjuvant chemotherapy could decrease the presence of micrometastases of cancer cells.

High heterogeneity existed in evaluating the RFS and OS rate between the chemotherapy and observation group. Given that heterogeneity in some analyses may impose a distinct effect on the synthesis results, the mechanisms behind it deserve to be figured out. More importantly, this heterogeneity may implicate the optimal subgroup of patients who could obtain the clinical benefits from postoperative adjuvant chemotherapy. As the JCOG9304[23] indicated, postoperative cisplatin with vindesine chemotherapy did not show efficacious in cases of resected N2 NSCLC. Meanwhile, the JLCSSG trial also failed to elucidate the therapeutic benefits of postoperative cisplatin plus vindesine chemotherapy, since the eligible patients in the trial were limited to be stage III.[25] Therefore, it is reasonable to infer that patients with stage III (T3 or any N2) may derive very limited benefits from the adjuvant 'cisplatin_vindesine'.

According to the subgroup analysis conducted by LACE in 2010, 'cisplatin_vinorelbine' was an effective therapeutic method for the improvement of survival with a manageable toxicity profile.[8] Over the few decades, the introduction of the other third-generation cytotoxicity chemotherapy regimens such as pemetrexed and docetaxel have brought into the development of platinum adjuvant chemotherapy.[10–12] Noteworthy, regardless of the absence of level one data in the adjuvant setting, due to its gold status in non-squamous advanced NSCLC with superior efficacy and low tolerability, 'cisplatin_pemetrexed' is still reasonably recommended as a standard of care for patients with early-stage resectable non-squamous NSCLC.[13 14] However, our results indicated that 'cisplatin_pemetrexed' did not show its significant survival advantages over the observation arm in the adjuvant setting. In addition, a subgroup analysis conducted by a large clinical trial-ECOG1505, did not find a significant difference among the four platinum-based chemotherapy regimens : 'cisplatin_vinorelbine', 'cisplatin_docetaxel', 'cisplatin_gemcitabine' and 'cisplatin_pemetrexed'.[46] Of note, the absolute effect of adjuvant cytotoxicity chemotherapy on stage IB and squamous patients is still under debated.[10 47] Hence, it seemed that the 'cisplatin_vinorelbine' arm still played a vital role in the adjuvant setting for patients with NSCLC since its efficacy was not influenced by the histology or stage.[8] By contrast, the negative results of 'cisplatin_pemetrexed' implies the restrictions of its application in the adjuvant chemotherapy setting for patients with squamous. Moreover, some biomarkers such as the excision repair cross-complementation group 1, b-tubulin and mucin hold the potential to predict the efficacy of cytotoxicity chemotherapy in the adjuvant setting.[48–50] Therefore, the heterogeneity of biological characteristics of each patient imposed confounding factors to the comparisons of different chemotherapy regimens.

According to our analysis, 'cisplatin_etoposide', the gold chemotherapy regimens for small cell lung cancer,

did have a significantly higher survival rate over observation arm in the patients with early-stage resectable NSCLC. Roselli *et al* conducted a 10-year follow-up clinical trials to demonstrate the effect of 'cisplatin_etoposide' on the reduction of recurrences and the extension of long-term survival, which enrolled only patients with stage I NSCLC and adopted the relatively high dose of cisplatin ($100\,mg/m^2$) and etoposide ($120\,mg/m^2$).[34] Hence, considering the potential bias, the positive results of 'cisplatin_etoposide' need to be interpreted with caution.

Antiangiogenic therapy, such as bevacizumab and endostar, hold the potential to improve the efficacy of adjuvant chemotherapy probably through inhibiting the growth of micrometastatic cancer cells and increasing the sensitivity of tumour cells to chemotherapy.[27 51 52] However, our results showed that the combination of chemotherapy regimens and endostar did not show advantage over a single platinum-based chemotherapy arm. Furthermore, according to the results of ECOG1505 trial, there was no significant difference between the adjuvant chemotherapy with or without bevacizumab for patients with resected NSCLC.[46] One of the reasons to explain these negative results might be the heterogeneity in the expression of vascular endothelial growth factor (VEGF). Patients with high VEGF expression tend to obtain more benefits from antiangiogenic therapy than the low VEGF expression ones.[27]

In terms of the safety of adjuvant chemotherapy, the incidence of haematological toxicities and nausea/vomiting, which was greatly influenced by the dose and schedules, were not significantly higher in the 'cisplatin_vinorelbine' arm than other chemotherapy. Noteworthy, the schedules in the adjuvant setting was quite different from the one utilised in the advanced NSCLC patients.[53] Therefore, 'cisplatin_vinorelbine' could be recommended as an adjuvant cytotoxicity chemotherapy regimen with an acceptable safety profile. On the contrary, the incidence of anaemia and thrombocytopaenia in the 'carboplatin_pemetrexed' group was significantly higher than the 'cisplatin_vinorelbine' group. Both the 'carboplatin' and 'pemetrexed' are characterised as relatively safe cytotoxicity chemotherapy regimens and recommended to be utilised for patients with medical commodities. The contradictory outcomes could be explained in part by the relatively lower dose of 'cisplatin_vinorelbine'.

To the best of our knowledge, this study was the first to directly compare the efficacy and safety of different adjuvant chemotherapy regimens for NSCLC patients. Despite the strengths, it was evident that there still existed some limitations in this study. First, for nearly half of the enrolled studies, the risk of bias in certain items remained unclear and the quality of them could not be evaluated. Secondary, due to the lack of specific data, the subgroup analysis stratified by stage and pathology could not be performed. Thus, it is difficult to determine the optimal subgroup populations who may obtain the most benefits from certain cytotoxicity chemotherapy regimens. Furthermore, we did not incorporate any targeted or immunebiological therapy into the analysis due to the open-loop of interventions. Last but not at least, although the absolute effect of radiotherapy for patients with completely resected NSCLC was presented recently, we did not conduct a subgroup analysis stratified by postoperative radiotherapy or not as it was undertaken according to every centre's policy.

## CONCLUSION

In conclusion, this study summarised the adjuvant cytotoxicity chemotherapy regimens for patients with early-stage resected NSCLC. Research on adjuvant cytotoxicity chemotherapy might be an out-of-date topic but numerous NSCLC patients could obtain benefit from the optimal cytotoxicity chemotherapy regimen. 'Cisplatin_vinorelbine' had a significant survival advantage with a relatively good safety profile in the adjuvant setting while the 'cisplatin_pemetrexed' arm was not superior to the other therapeutic methods in improving survival. Our analysis may provide some guiding significance for the clinicians when determining the optimal chemotherapy regimen.

**Acknowledgements** This work was financially supported by the Chinese National Natural Science Foundation Project (81772476). We thank all the co-authors for their contributions to this study.

**Contributors** W-FF: conceptualisation; L-LP, JG and Y-HH: Data curation; L-LP and J-DG: Formal analysis; W-FF: Funding acquisition; L-LP: Investigation; L-LP: Methodology; W-FF: Project administration; L-LP: Resources; L-LP: Software, LZ and W-FF: Supervision; J-DG: Validation; L-LP: Visualisation; All authors, L-LP, J-DG, Y-HH, JL, YL, WA-SA, LZ, W-FF: Roles/writing-original draft; all authors L-LP, J-DG, Y-HH, JL, YL, WA-SA, LZ and W-FF: writing-review and editing. W-FF is responsible for the overall content as guarantor.

**Funding** This work was supported by the Chinese National Natural Science Foundation Project (81772476).

**Competing interests** None declared.

**Patient and public involvement** Patients and/or the public were not involved in the design, or conduct, or reporting, or dissemination plans of this research.

**Patient consent for publication** Not applicable.

**Provenance and peer review** Not commissioned; externally peer reviewed.

**Data availability statement** No data are available. Data are available in a public, open access repository. All data relevant to the study are included in the article or uploaded as online supplementary information.

**ORCID iD**
Lan-Lan Pang http://orcid.org/0000-0002-4932-8117

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
