## [Reviewer comments · BMJ Open]

ARTICLE DETAILS

TITLE (PROVISIONAL)	Investigation of the Optimal Platinum-based Regimen in the Postoperative Adjuvant Chemotherapy Setting for Early-stage Resected Non-small Lung Cancer: A Bayesian Network Meta-analysis
AUTHORS	Pang, Lanlan; Gan, Jiadi; Huang, Yi-Hua; Liao, Jun; Lv, Yi; Ali, Wael; Zhang, L; Fang, Wenfeng;

VERSION 1 – REVIEW

REVIEWER	Yano, Yoshitaka Kyoto Pharmaceutical University, Education and Research
REVIEW RETURNED	11-Oct-2021

GENERAL COMMENTS	The manuscript describes the results of a Bayesian network meta-analysis (NMA) to examine efficacy and safety of the platinum-based regimens for postoperative therapy of NSCLC. The analysis seems to be appropriately performed according to the general procedures of NMA. I have some minor comments as follows: 1) Running title: The findings were obtained by the NMA, so a term “meta-analysis” can be included in the running title.2) Line 246: Explanations of Figure 3A seems not enough. Please add some sentences what was found from Figure 3A.3) Line 286: Explanations of Figure 5A seems not enough. Please add some sentences what was found from Figure 5A. (same comment as 2) above).4) Line 257: The text says “there were no significant differences among the regimens .. (Figure 3C)”, but a cell in Figure 3C highlighted in green shows the significant confidence interval (upper level is less than 1.0). Please clarify the consistency between the text and the Figure.5) Page 42: Titles – Figure 4.B, 4.C, 4.D should be Figure 3.B., 3.C., 3.D, respectively.6) Line 296: The text says “there were no significant differences among the regimens .. (Figure 5C)”, but a green cell in Figure 5C shows a significant confidence interval. Please clarify (same comment as 4) above).7) Page 45: numbers of x-axes in Figures 6A – 6E are difficult to understand the scale – they do not look good. Isn't it possible to use “0” and “+infinite” instead of these numbers, and to add some sub-scales between 1 and them?
---

REVIEWER	Burdett, Sarah MRC Clinical Trials Unit at UCL, 2nd Floor
REVIEW RETURNED	20-Jan-2022

GENERAL COMMENTS	Thank you for the opportunity to review this paper. I am commenting only on the systematic review aspect of this paper not the meta-analysis and associated statistics. 3. Is the study design appropriate to answer the research question? 4. Are the methods described sufficiently to allow the study to be repeated? 6. Are the outcomes clearly defined? I have answered 'no' to these questions as I have a number of concerns. I am very familiar with the set of trials comparing platinum vs observation but less so for the platinum vs platinum set of trials, so I can only comment of the platinum vs observation trials. Eligibility criteria It is unclear from the eligibility criteria whether only trials using Vinorelbine, Etoposide, Pemetrexed, Docetaxel, Paclitaxel, Gemcitabine and Vindesine were considered eligible or if this was all that was found. I am aware of a further 5 RCTs that include etoposide and 4 RCTs that use other chemotherapies in combination with platinum. It may be that there was not enough information in the publications of the trials to be used here, but the etoposide trials at least are eligible and if not used, there should be an explanation why not. I have not looked in detail at the chemo vs chemo set of trials. Definition of outcomes OS and RFS have been defined as time from surgery to death (OS) or recurrence or death (RFS). I would be surprised if time from surgery was reported in these trials as I think most reported outcomes from the time of randomisation. If the authors have got this information from those who carried out the trials, this should definitely be mentioned, otherwise the definitions should be looked at again. Risk of bias Risk of bias in the trials has been assessed but it is not referenced. It is not clear which version of the ROB assessment has been used. ROB2 is increasingly used. https://methods.cochrane.org/bias/resources/rob-2-revised-cochrane-risk-bias-tool-randomized-trials 7. If statistics are used are they appropriate and described fully? I have answered N/A to this question as this needs a statistical review. 8. Are the references up-to-date and appropriate? As mentioned before, ROB is not referenced in the paper and although the authors include reference 37 (Non-small Cell Lung Cancer Collaborative Group (1995). "Chemotherapy in non-small cell lung cancer: a meta-analysis using updated data on individual patients from 52 randomised clinical trials." BMJ 311: 899-909.) I would suggest they also include the update of this work published in 2010 (NSCLC Meta-analyses Collaborative Group (2010). "Adjuvant chemotherapy, with or without postoperative radiotherapy, in operable non-small cell lung cancer: two meta-analyses of individual patient data." Lancet 375(9722): 1267-1277.) This includes references to the trials not included in this paper.
--

REVIEWER	Ramos-Esquivel , A Departamento de Oncología Médica, Hospital San Juan de Dios, Universidad de Costa Ric
REVIEW RETURNED	03-Feb-2022

GENERAL COMMENTS	The authors performed a comprehensive analysis about this interesting topic. The research question, systematic review and the statistical methods were well described and addressed. My major comment is related to the high heterogeneity found in some analyses (Figure 4 C and D) that is not addressed in the Discussion section. I recommend to point out the source of heterogeneity in trials like JCOG9304, and the one published in J THORAC CARDIOVASC SURG 1993;106:703-8. Minor comments:  1. Please change the numerical scale in Figure 6 (A to E) 2. Please add a "," before the year of each trial, and try to identify all trials by the name of the author of the name of the trial but not both. 3. Please describe what is meant by SUCRA, since not all readers are familiar with this term. 4. Please do not use abbreviations or acronyms in figures or titles (i.e., OS, RFS, OR, 95%CrI, and 95%CI) or explain them.. 5. In some parts of the text the authors refer to each trial in terms of the name of it or the authors' name. Please use the same method throughout the text.
---

VERSION 1 – AUTHOR RESPONSE

Reviewer: 1

Prof. Yoshitaka Yano, Kyoto Pharmaceutical University

Comments to the Author:

The manuscript describes the results of a Bayesian network meta-analysis (NMA) to examine the efficacy and safety of the platinum-based regimens for postoperative therapy of NSCLC. The analysis seems to be appropriately performed according to the general procedures of NMA. I have some minor comments.

Re: Thanks for your encouraging review. We have modified the manuscript and figures according to your enlightening and constructive suggestions.

1) Running title: The findings were obtained by the NMA, so a term “meta-analysis” can be included in the running title.

Re: Thanks for your advice. We have added the term---“meta-analysis” in the running title to enable the readers to have a more objective understanding of the present work (Page 1).

2) Line 246: Explanations of Figure 3A seems not enough. Please add some sentences what was found from Figure 3A.

3) Line 286: Explanations of Figure 5A seems not enough. Please add some sentences what was found from Figure 5A. (same comment as 2) above).

Re: We appreciate your comments. In the Network evidence figure, each node represents a type of treatment. Solid lines connect treatments that are directly compared in at least one study. The thickness of connections varies according to the number of studies involved in a comparison. We have replenished a notation concerning the Network evidence figure in the revised figure legends (Page 35). Moreover, some findings obtained from Figure 3 A and Figure 5A have been indicated in the results part (Page 15, 17).

4) Line 257: The text says “there were no significant differences among the regimens. (Figure 3C)”, but a cell in Figure 3C highlighted in green shows the significant confidence interval (upper level is less than 1.0). Please clarify the consistency between the text and the Figure.

Re: We are sorry that we did not make the interpretation clearly understood. A cell in Figure 3C that we highlighted in green did show a significant confidence interval. However, this statistical significance existed in the comparison between the observation and “Cisplatin_Vinorelbine”; whereas no significant difference was observed in the remaining comparisons, eg “Cisplatin_Vinorelbine” and “Cisplatin_Pemetrexmed”. The significant difference between the observation and “Cisplatin_Vinorelbine” has been mentioned above. Conversely, there was no significant difference among these adjuvant platinum-based chemotherapy regimens. We have modified the related sentences in the revised manuscript (Page 15,17).

5) Page 42: Titles – Figure 4.B, 4.C, 4.D should be Figure 3.B., 3.C., 3.D, respectively.

Re: Thank you for bringing this mistake to our attention. We have corrected it in the revised Figure 4.

6) Line 296: The text says “there were no significant differences among the regimens .. (Figure 5C)”, but a green cell in Figure 5C shows a significant confidence interval. Please clarify (same comment as 4) above).

Re: Thanks for your advice again. We have adjusted the related statement in the revised text (Page 17).

7) Page 45: numbers of x-axes in Figures 6A – 6E are difficult to understand the scale – they do not look good. Isn't it possible to use “0” and “+infinite” instead of these numbers, and to add some sub-scales between 1 and them?

Re: Thanks for your enlightening recommendations. We have modified the numerical scale of x-axes in Figure 6A-E. After normalized by logs base e (Napierian logarithm), the appearance in Figure A-E was shown in an appropriate way.

Reviewer: 2

Dr. Sarah Burdett, MRC Clinical Trials Unit at UCL

Comments to the Author

I am very familiar with the set of trials comparing platinum vs observation but less so for the platinum vs platinum set of trials, so I can only comment of the platinum vs observation trials.

Eligibility criteria

It is unclear from the eligibility criteria whether only trials using Vinorelbine, Etoposide, Pemetrexed, Docetaxel, Paclitaxel, Gemcitabine and Vindesine were considered eligible or if this was all that was found. I am aware of a further 5 RCTs that include etoposide and 4 RCTs that use other chemotherapies in combination with platinum. It may be that there was not enough information in the publications of the trials to be used here, but the etoposide trials at least are eligible and if not used,

there should be an explanation why not. I have not looked in detail at the chemo vs chemo set of trials.

Re: It is our pleasure to receive your comments on the present work. The reason for “Vinorelbine, Etoposide, Pemetrexed, Docetaxel, Paclitaxel, Gemcitabine and Vindesine” as the eligible therapeutic regimens were the fact that these above-mentioned drugs are currently commonly used in the routine clinical practice (Page 10). Furthermore, head-to-head clinical trials concerning adjuvant chemotherapy mainly focus on these therapeutic regimens in combination with Cisplatin or Carboplatin. The patients in the present study were restrained to be early-stage NSCLC. Hence, RCTs that focused on SCLC were not enrolled in the meta-analysis. Of note, the remaining four RCTs including vinca alkaloid/etoposide finally were not enrolled in this meta-analysis[1-4]. Actually, we have noticed these four well-designed RCTs during the process of eligible trial selection. Nevertheless, we could not obtain the specific outcomes in the subgroup populations per treatment regimens from these clinical trials, whereas only the outcomes concerning the comparison between the observation group and “Cisplatin_Vinorelbine” were available in BLT or IALT. Given that the objective of the present work was to investigate the optimal platinum-based regimen in the postoperative adjuvant chemotherapy setting for early-stage resected NSCLC, we only enrolled the BLT and IALT while leaving the remaining four RCTs out. We have added an explanation for the unavailable information in these four RCTs and their exclusion reasons (Page 14).

Reference:

1. Park, J.H., et al., Postoperative adjuvant chemotherapy for stage I non-small cell lung cancer☆. *European Journal of Cardio-Thoracic Surgery*, 2005. 27(6): p. 1086-1091.
2. Park, J.H., P2-202: Postoperative adjuvant therapy for stage IIIA non small cell lung cancer. *Journal of Thoracic Oncology*, 2007. 2(8, Supplement): p. S651.
3. Scagliotti, G.V., et al., Randomized study of adjuvant chemotherapy for completely resected stage I, II, or IIIA non-small-cell Lung cancer. *J Natl Cancer Inst*, 2003. 95(19): p. 1453-61.
4. A randomized controlled study of postoperative adjuvant chemoimmunotherapy of resected non-small cell lung cancer with IL2 and LAK cells: H.Kimura, Y.Yamaguchi, T.Fujisawa, M.Baba, and M.Shiba Institute of Pulmonary Cancer Research, School of Medicine, Chi %J Lung Cancer. 1991.

Definition of outcomes

OS and RFS have been defined as time from surgery to death (OS) or recurrence or death (RFS). I would be surprised if time from surgery was reported in these trials as I think most reported outcomes from the time of randomisation. If the authors have got this information from those who carried out the trials, this should definitely be mentioned, otherwise the definitions should be looked at again.

Re: Thanks a lot for bringing this to our attention. Indeed, most enrolled RCTs in this meta-analysis defined the OS or RFS from randomized assignment to death (OS) or recurrence or death (RFS) except for three clinical trials. After careful deliberation, we decided to correct the definition of OS and RFS. We completely agreed with your comments. RFS should be defined as the interval from randomization to disease recurrence or death, whichever occurred first, while OS refers to the time from randomization to death due to any cause (Page 10).

Risk of bias

Risk of bias in the trials has been assessed but it is not referenced. It is not clear which version of the ROB assessment has been used. ROB2 is increasing used.

<https://methods.cochrane.org/bias/resources/rob-2-revised-cochrane-risk-bias-tool-randomized-trials>.

Re: We appreciate your comments. The ROB assessment utilized in the present study was Version 5 of the Cochrane Handbook updated in 2011[5]. We have replenished the related reference in the revised manuscript (Page 11).

Reference:

5. Barger, C.J., et al., Pan-Cancer Analyses Reveal Genomic Features of FOXM1 Overexpression in Cancer. *Cancers (Basel)*, 2019. 11(2).

As mentioned before, ROB is not referenced in the paper and although the authors include reference 37 (Non-small Cell Lung Cancer Collaborative Group (1995). "Chemotherapy in non-small cell lung cancer: a meta-analysis using updated data on individual patients from 52 randomised clinical trials." *BMJ* 311: 899-909.)

I would suggest they also include the update of this work published in 2010 (NSCLC Meta-analyses Collaborative Group (2010). "Adjuvant chemotherapy, with or without postoperative radiotherapy, in operable non-small cell lung cancer: two meta-analyses of individual patient data." *Lancet* 375(9722): 1267-1277.) This includes references to the trials not included in this paper.

Re: Thanks a lot for your suggestions. We have added the update of this work published in 2010 into the references list. We hope it will help readers to better understand the research progress in the adjuvant chemotherapy field (Page 21).

Reviewer: 3

Dr. A Ramos-Esquivel, Departamento de Oncología Médica, Hospital San Juan de Dios, Universidad de Costa Ric

Comments to the Author:

The authors performed a comprehensive analysis about this interesting topic. The research question, systematic review and the statistical methods were well described and addressed.

Re: Thank you for this encouraging review. We have revised the manuscript accordingly to make the present study much improved.

My major comment is related to the high heterogeneity found in some analyses (Figure 4 C and D) that is not addressed in the Discussion section. I recommend to point out the source of heterogeneity in trials like JCOG9304, and the one published in *J THORAC CARDIOVASC SURG* 1993;106:703-8.

Re: Thanks for your constructive views on this manuscript. Given that the high heterogeneity in some analyses may impose a distinct effect on the synthesis results, the reasons for high heterogeneity deserve to be figured out. And more importantly, the heterogeneity may provide a guiding significance for the optimal subgroup of patients who may obtain the clinical benefits from postoperative adjuvant chemotherapy. In JCOG9304 trial, postoperative cisplatin with vindesine chemotherapy was not shown to be efficacious in cases of resected N2 non-small cell lung cancer. Meanwhile, the trial published in *J THORAC CARDIOVASC SURG* 1993 also failed to demonstrate the therapeutic benefits of postoperative cisplatin and vindesine chemotherapy, since the eligible patients in the trial were restrained to be stage III. It is reasonable to assume that the benefits of adjuvant "Cisplatin_Vindesine" may be limited in patients with stage III (T3 or any N2). We have replenished the related interpretation in the revised discussion part (Page 22, 23).

Minor comments:

1. Please change the numerical scale in Figure 6 (A to E)

Re: Thanks for your enlightening recommendations. We have modified the numerical scale of x-axes in Figure 6A-E. After normalized by logs base e (Napierian logarithm), the appearance in Figure A-E was shown in an appropriate way.

2. Please add a "," before the year of each trial, and try to identify all trials by the name of the author of the name of the trial but not both.

Re: Thanks for bringing this to our attention. We have added a "," before the year of each trial in the revised figures. Trials that were presented in the form of original names may be more representative and impactful than that of the first author's name. Nevertheless, given that not all trials owned their unique trial name during the study design process, we could not identify all trials by the name of the trial. Furthermore, some previously well-designed published meta-analyses also referred to the trial in the form of both the name of trial and author name [6, 7]. Hence, after thoughtful consideration, we tried to identify the enrolled trials by the name of the trial except for several trials in the present study.

Reference:

6. Arriagada, R., et al., Adjuvant chemotherapy, with or without postoperative radiotherapy, in operable non-small-cell lung cancer: two meta-analyses of individual patient data. *Lancet*, 2010. 375(9722): p. 1267-77.

7. Chemotherapy in non-small cell lung cancer: a meta-analysis using updated data on individual patients from 52 randomised clinical trials. Non-small Cell Lung Cancer Collaborative Group. *Bmj*, 1995. 311(7010): p. 899-909.

3. Please describe what is meant by SUCRA, since not all readers are familiar with this term.

Re: We appreciate your comments. The surface under the cumulative ranking curve (SUCRA), represents the percentage of efficacy achieved by an agent compared to an imaginary agent that is always the best without uncertainty (ie, SUCRA = 100%)[8]. Namely, the SUCRA value would be 0 if treatment is certain to be the worst and 1 if it is certain to be the best. Higher SUCRA scores correspond to a higher ranking for extending survival. We have replenished the notation of SUCRA in the methods section (Page 12).

Reference:

8. Salanti, G., A.E. Ades, and J.P. Ioannidis, Graphical methods and numerical summaries for presenting results from multiple-treatment meta-analysis: an overview and tutorial. *J Clin Epidemiol*, 2011. 64(2): p. 163-71.

4. Please do not use abbreviations or acronyms in figures or titles (i.e., OS, RFS, OR, 95%CrI, and 95%CI) or explain them.

Re: Thank you for bringing this to our attention. We have removed all the abbreviations or acronyms in the related figures legends and replaced them with corresponding full names.

5. In some parts of the text the authors refer to each trial in terms of the name of it or the authors' name. Please use the same method throughout the text.

Re: Thanks for your advice again. We have tried to identify the enrolled trials by the name of the trial in the revised text.

VERSION 2 – REVIEW

REVIEWER	Burdett, Sarah MRC Clinical Trials Unit at UCL, 2nd Floor
REVIEW RETURNED	16-Mar-2022

GENERAL COMMENTS	Thank you, all of the comments I originally made have been addressed by the authors. Some of the new text in the discussion and conclusions (particularly last sentence of conclusion), need to be reworded for clarity. One thing I would like to ask the authors is regarding the definitions of OS and RFS. They originally from date of surgery to date of event, and now they are date of randomisation to date of event. Was 'surgery' an error originally, or have the results been changed? I can't tell as the figures aren't included in this version of the manuscript.
--

VERSION 2 – AUTHOR RESPONSE

Reviewer: 2

Some of the new text in the discussion and conclusions (particularly last sentence of conclusion), need to be reworded for clarity.

Re: Thank you for your suggestions. We have reworded the new text in the discussion and conclusions. In doing so, the related statement is much improved.

One thing I would like to ask the authors is regarding the definitions of OS and RFS. They originally from date of surgery to date of event, and now they are date of randomisation to date of event. Was 'surgery' an error originally, or have the results been changed? I can't tell as the figures aren't included in this version of the manuscript.

Re: Thanks for bringing this to our attention. Most enrolled RCTs in our study defined the OS or RFS from randomized assignment to death (OS) or recurrence or death (RFS) except for three clinical trials[1-3]. By contrast, the remaining three RCTs defined the RFS as “the time from surgical resection to either relapse or death, whichever came first”. It is reasonable to infer that the time interval from surgery to randomization was relatively short or even the same day in their studies. All the survival data in this meta-analysis was directly obtained from what the enrolled RCTs reported. Consequently, the results were not influenced by the correction of defining OS or RFS.

Thanks again for your time on this manuscript.

References:

1. Kreuter, M., et al., Three-Year Follow-Up of a Randomized Phase II Trial on Refinement of Early-Stage NSCLC Adjuvant Chemotherapy with Cisplatin and Pemetrexed versus Cisplatin and Vinorelbine (the TREAT Study). *J Thorac Oncol*, 2016. 11(1): p. 85-93.

2. Kubota, K., et al., Randomized phase II trial of adjuvant chemotherapy with docetaxel plus cisplatin versus paclitaxel plus carboplatin in patients with completely resected non-small cell lung cancer: TORG 0503. *Lung Cancer*, 2020. 141: p. 32-36.
3. Wang, J., et al., Post-operative treatment with cisplatin and vinorelbine in Chinese patients with non-small cell lung cancer: a clinical prospective analysis of 451 patients. *Asian Pac J Cancer Prev*, 2012. 13(9): p. 4505-10.